# GraphPNAS: Learning Distributions of Good Neural Architectures via Deep Graph Generative Models

## Abstract

Neural architectures can be naturally viewed as computational graphs. Motivated by this perspective, we, in this paper, study neural architecture search (NAS) through the lens of learning random graph models. In contrast to existing NAS methods which largely focus on searching for a single best architecture, *i.e.*, point estimation, we propose *GraphPNAS*, a deep graph generative model that learns a distribution of well-performing architectures. Relying on graph neural networks (GNNs), our GraphPNAS can better capture topologies of good neural architectures and relations between operators therein. Moreover, our graph generator leads to a learnable probabilistic search method that is more flexible and efficient than the commonly used RNN generator and random search methods. Finally, we learn our generator via an efficient reinforcement learning formulation for NAS. To assess the effectiveness of our GraphPNAS, we conduct extensive experiments on three search spaces, including the challenging RandWire on Tiny-ImageNet, ENAS on CIFAR10, and NAS-Bench-101. The complexity of RandWire is significantly larger than other search spaces in the literature. We show that our proposed graph generator consistently outperforms RNN based one and achieves better or comparable performances than state-of-the-art NAS methods.

## 1 Introduction

In recent years, we have witnessed a rapidly growing list of successful neural architectures that underpin deep learning, *e.g.*, VGG, LeNet, ResNets (He et al., 2016), Transformers (Dosovitskiy et al., 2020). Designing these architectures requires researchers to go through time-consuming trial and errors. Neural architecture search (NAS) (Zoph & Le, 2016; Elsken et al., 2018b) has emerged as an increasingly popular research area which aims to automatically find state-of-the-art neural architectures without human-in-the-loop.

NAS methods typically have two components: a search module and an evaluation module. The search module is expressed by a machine learning model, such as a deep neural network, designed to operate in a high dimensional search space. The search space, of all admissible architectures, is often designed by hand in advance. The evaluation module takes an architecture as input and outputs the reward, *e.g.*, performance of this architecture trained and then evaluated with a metric. The learning process of NAS methods typically iterates between the following two steps. 1) The search module produces candidate architectures and sends them to the evaluation module; 2) The evaluation module evaluates these architectures to get the reward and sends the reward back to the search module. Ideally, based on the feedback from the evaluation module, the search module should learn to produce better and better architectures. Unsurprisingly, this learning paradigm of NAS methods fits well to reinforcement learning (RL).

Most NAS methods (Liu et al., 2018b; White et al., 2020; Cai et al., 2019) only return a single best architecture (*i.e.*, a point estimate) after the learning process. This point estimate could be very biased as it typically underexplores the search space. Further, a given search space may contain multiple (equally) good architectures, a feature that a point estimate cannot capture. Even worse, since the learning problem of NAS is essentially a discrete optimization where multiple local minima exist, many local search style NAS methods (Ottelander et al., 2020) tend to get stuck in local minima. From the Bayesian perspective, modelling the distribution of architectures is inherently better than

point estimation, *e.g.*, leading to the ability to form ensemble methods that work better in practice. Moreover, modelling the distribution of architectures naturally caters to probabilistic search methods which are better suited for avoiding local optima, *e.g.*, simulated annealing. Finally, modeling the distribution of architectures allows to capture complex structural dependencies between operations that characterize good architectures capable of more efficient learning and generalization.

Motivated by the above observations and the fact that neural architectures can be naturally viewed as attributed graphs, we propose a probabilistic graph generator which models the distribution over good architectures using graph neural networks (GNNs). Our generator excels at generating topologies with complicated structural dependencies between operations. From the Bayesian inference perspective, our generator returns a distribution over good architectures, rather than a single point estimate, allowing to capture the multi-modal nature of the posterior distribution of good architectures and to effectively average or ensemble architecture (sample) estimates. Different from the Bayesian deep learning (Neal, 2012; Blundell et al., 2015; Gal & Ghahramani, 2016) that models distributions of weights/hidden units, we model distributions of neural architectures. Lastly, our probabilistic generator is less prone to the issue of local minima, since multiple random architectures are generated at each step during learning. In summary, our key contributions are as below.

- We propose a GNN-based graph generator for neural architectures which empowers a learnable probabilistic search method. To the best of our knowledge, we are the first to explore learning deep graph generative models as generators in NAS.

- We explore a significantly larger search space (*e.g.*, graphs with 32 operators) than the literature (*e.g.*, garphs with up to 12 operators) and propose to evaluate architectures under low-data regime, which altogether boost effectiveness and efficiency of our NAS system.

- Extensive experiments on three different search spaces show that our method consistently outperforms RNN-based generators and is slightly better or comparable to the state-of-the-art NAS methods. Also, it can generalize well across different NAS system setups.

## 2 RELATED WORKS

**Neural Architecture Search.** The main challenges in NAS are 1) the hardness of discrete optimization, 2) the high cost for evaluating neural networks, and 3) the lack of principles in the search space design. First, to tackle the discrete optimization, evolution strategies (ES) (Elsken et al., 2019; Real et al., 2019a), reinforcement learning (RL) (Baker et al., 2017; Zhong et al., 2018; Pham et al., 2018; Liu et al., 2018a), Bayesian optimization (Bergstra et al., 2013; White et al., 2019) and continuous relaxations (Liu et al., 2018b) have been explored in the literature. We follow the RL path as it is principled, flexible in injecting prior knowledge, achieves the state-of-the-art performances (Tan & Le, 2019), and can be naturally applied to our graph generator. Second, the evaluation requires training individual neural architectures which is notoriously time consuming(Zoph & Le, 2016). Pham et al. (2018); Liu et al. (2018b) propose a weight-sharing supernet to reduce the training time. Baker et al. (2018) use a machine learning model to predict the performance of fully-trained architectures conditioned on early-stage performances. Brock et al. (2018); Zhang et al. (2018) directly predict weights from the search architectures via hypernetworks. Since our graph generator do not relies on specific choice of evaluation method, we choose to experiment on both oracle training(training from scratch) and supernet settings for completeness. Third, the search space of NAS largely determines the optimization landscape and bounds the best-possible performance. It is obvious that the larger the search space is, the better the best-possible performance and the higher the search cost would likely be. Besides this trade-off, few principles are known about designing the search space. Previous work (Pham et al., 2018; Liu et al., 2018b; Ying et al., 2019; Li et al., 2020) mostly focuses on cell-based search space. A cell is defined as a small (*e.g.*, up to 8 operators) computational graph where nodes (*i.e.*, operators like 3×3 convolution) are connected following some topology. Once the search is done, one often stacks up multiple cells with the same topology but different weights to build the final neural network. Other works (Tan et al., 2019; Cai et al., 2019; Tan & Le, 2019) typically fix the topology, *e.g.*, a sequential backbone, and search for layer-wise configurations (*e.g.*, operator types like 3×3 vs. 5×5 convolution and number of filters). In our method, to demonstrate our graph generator's ability in exploring large topology search space, we first explore on a challenging large cell space (32 operators), after which we experiment on ENAS Macro (Pham et al., 2018) and NAS-Benchmark-101(Ying et al., 2019) for more comparison with previous methods.

**Neural Architecture as Graph for NAS.** Recently, a line of NAS research works propose to view neural architectures as graphs and encode them using graph neural networks (GNNs). In (Zhang et al., 2020; Luo et al., 2018), graph auto-encoders are used to map neural architectures to and back from a continuous space for gradient-based optimization. Shi et al. (2020) use bayesian optimization (BO), where GNNs are used to get embedding from neural architectures. Despite the extensive use of GNNs as encoders, few works focus on building graph generative models for NAS. Closely related to our work, Xie et al. (2019) explore different topologies of the similar cell space using non-learnable random graph models. You et al. (2020) subsequently investigate the relationship between topologies and performances. Following this, Ru et al. (2020) propose a hierarchical search space modeled by random graph generators and optimize hyper-parameters using BO. They are different from our work as we learn the graph generator to automatically explore the cell space.

**Deep Graph Generative Models.** Graph generative models date back to the Erdős–Rényi model (Erdös & Rényi, 1959), of which the probability of generating individual edges is the same. Other well-known graph generative models include the stochastic block model (Holland et al., 1983), the small-world model (Watts & Strogatz, 1998), and the preferential attachment model (Barabási & Albert, 1999). Recently, deep graph generative models instead parameterize the probability of generating edges and nodes using deep neural networks in, *e.g.*, the auto-regressive fashion (Li et al., 2018; You et al., 2018; Liao et al., 2019) or variational autoencoder fashion (Kipf & Welling, 2016; Grover et al., 2018; Liu et al., 2019). These models are highly flexible and can model complicated distributions of real-world graphs, *e.g.*, molecules (Jin et al., 2018), road networks (Chu et al., 2019), and program structures (Brockschmidt et al., 2018). Our graph generator builds on top of the state-of-the-art deep graph generative model in (Liao et al., 2019) with several important distinctions. First, instead of only generating nodes and edges, we also generate node attributes (*e.g.*, operator types in neural architectures). Second, since good neural architectures are actually latent, our learning objective maximizes the expected reward (*e.g.*, validation accuracies) rather than the simple log likelihood, thus being more challenging.

## 3 METHODS

The architecture of any feedforward neural network can be naturally represented as a directed acyclic graph (DAG), a.k.a., *computational graph*. There exist two equivalent ways to define the computational graph. First, we denote operations (*e.g.*, convolutions) as nodes and denote operands (*e.g.*, tensors) as edges which indicate how the computation flows. Second, we denote operands as nodes and denote operators as edges. We adopt the first view. In particular, a neural network $\mathcal{G}$ with $N$ operations is defined as a tuple $(A, X)$ where $A \in \{0,1\}^{N \times N}$ is an $N \times N$ adjacent matrix with $A_{ij} = 1$ indicates that the output of the $j$-th operator is used as the input of the $i$-th operator. For operator with multiple inputs, inputs are combined together (*e.g.*, using `sum` or `average` operator) before sending into the operator. $X$ is a $N$-size *attribute* vector encoding operation types. For any operation $i$, its operation type $X_i$ can only choose from a pre-defined list with length $D$, *e.g.*, $1 \times 1$, $3 \times 3$ or $5 \times 5$ convolutions. Note that for any valid feedforward architecture, $\mathcal{G}$ can not have loops. One sufficient condition to satisfy the requirement is to constrain $A$ to be a lower triangular matrix with zero diagonal (*i.e.*, excluding self-loops). This formalism creates a search space of $D^N 2^{N(N-1)/2}$ possible architectures, which is huge even for moderately large number of operators $N$ and number of operation types $D$. The goal of NAS is to find an architecture or a set of architectures within this search space that would perform well. For practical consideration, we search for cell graphs (*e.g.*, $N = 32$) and then replicate this cell several times to build a deep neural architecture. We also experiment on the ENAS Macro search space where $\mathcal{G}$ defines a entire network. More details for the corresponding search spaces can be found in Section 4.

### 3.1 NEURAL ARCHITECTURE SEARCH SYSTEM

Before delving into details, we first give an overview of our NAS system, which consists of two parts: a generator and an evaluator. The system diagram is shown in Fig. 1. At each step, the probabilistic graph generator samples a set of cell graphs, which are further translated to neural architectures by replicating the cell graph multiple times and stacking them up. Then the evaluator evaluates these architectures, obtains rewards, and sends architecture-reward pairs to the replay buffer. The replay buffer is then used to improve the generator, effectively forming a reinforcement learning loop.

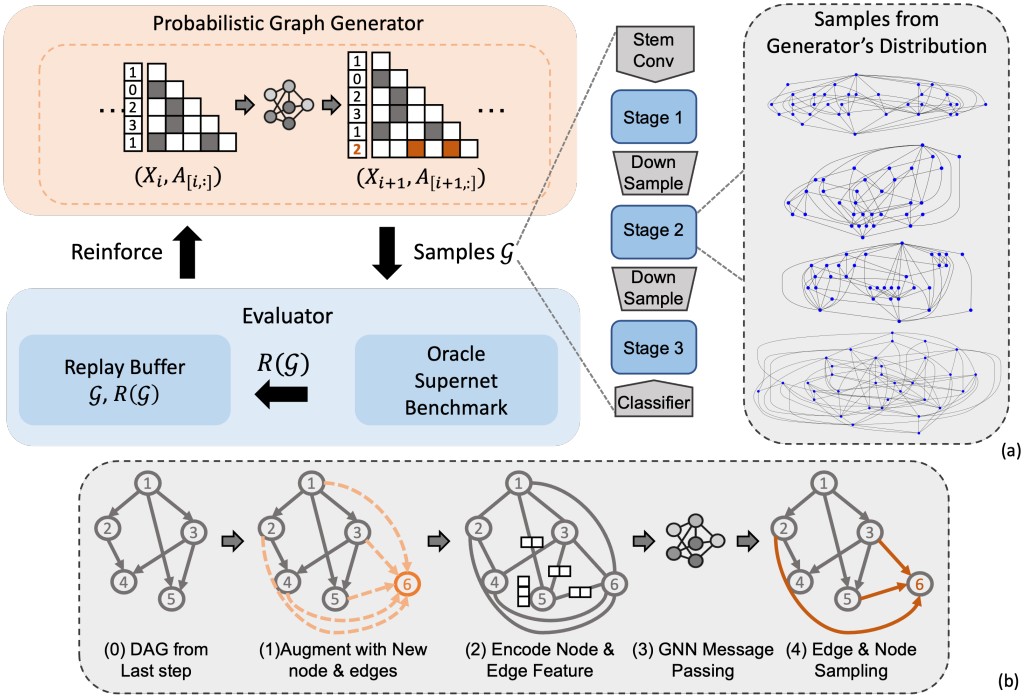

Figure 1: Figure (a) is the pipeline of our NAS system. The core part is a GNN-based graph generator from which we sample graph representations of neural network $\mathcal{G}$. The corresponding model for each $\mathcal{G}$ is then sent to evaluator for evaluation. The evaluation result is first stored in a replay buffer and then used for learning the graph generator through Reinforcement Learning. Figure(b) shows one generation step in the proposed probabilistic graph generator.

### 3.1.1 PROBABILISTIC GENERATORS FOR NEURAL ARCHITECTURES

Now we introduce our probabilistic graph generator which is based on a state-of-the-art deep auto-regressive graph generative model in (Liao et al., 2019).

**Auto-Regressive Generation.** Specifically, we decompose the distribution of a cell graph along with attributes (operation types) in an auto-regressive fashion,

$$\mathbb{P}(A,X) = \prod_{i=1}^{N} \mathbb{P}\left(A_{i,:}|A_{i-1,:}, X_{i-1}, \cdots, A_{1,:}, X_1\right) \mathbb{P}\left(X_i|A_{i-1,:}, X_{i-1}, \cdots, A_{1,:}, X_1\right), \quad (1)$$

where $A_{i,:}$ and $X_i$ denote the $i$-th row of the adjacency matrix $A$ and the $i$-th operation type respectively. To ensure the generated graphs are DAGs, we constrain $A$ to be lower triangular by adding a binary mask, *i.e.*, the $i$-th node can only be reached from the first $i-1$ nodes. We omit the masks in the equations for better readability. We further model the conditional distributions as follows,

$$\mathbb{P}\left(A_{i,:}|A_{i-1,:}, X_{i-1}, \cdots, A_{1,:}, X_1\right) = \sum_{k=1}^{K} \alpha_k \prod_{1 \le j < i} \theta_{k,i,j} \quad (2)$$

$$\mathbb{P}\left(X_i|A_{i-1,:}, X_{i-1}, \cdots, A_{1,:}, X_1\right) = \mathrm{Categorical}\left(\beta_1, \cdots, \beta_D\right) \quad (3)$$

$$\alpha_1, \ldots, \alpha_K = \mathrm{Softmax}\left(\sum\nolimits_{1 \le j < i} \mathrm{MLP}_\alpha(h_i^S - h_j^S)\right) \quad (4)$$

$$\beta_1, \ldots, \beta_D = \mathrm{Softmax}\left(\mathrm{MLP}_\beta(h_i^S)\right) \quad (5)$$

$$\theta_{1,i,j}, \ldots, \theta_{K,i,j} = \mathrm{Sigmoid}\left(\mathrm{MLP}_\theta(h_i^S - h_j^S)\right), \quad (6)$$

where the distributions of the operation type and edges are categorical and $K$-mixture of Bernoulli respectively. $D$ is again the number of operation types. $\mathrm{MLP}_\alpha$, $\mathrm{MLP}_\beta$, and $\mathrm{MLP}_\theta$ are different instances of two-layer MLPs with ReLU activations. Here $h_i^S$ is the representation of $i$-th node

returned by a GNN which has been executed $S$ steps of message passing at each generation step. This auto-regressive construction breaks down the nice property of permutation invariance for graph generation. However, we do not find it as an issue in practice, partly due to the fact that the graph isomorphism becomes less likely to happen while considering both topology and operation types.

**Message Passing GNNs.** Each generation step $n \leq N$ in auto-regressive generation above relies on representations of nodes up to and including $n$ itself (see Eq. (4)–(6)). To obtain these node representations $\{h_i^S\}$, we exploit message passing GNNs (Gilmer et al., 2017) with an attention mechanism similar to (Liao et al., 2019). In particular, the $s$-th message passing step involves executing the following equations successively,

$$m_{ij}^s = f([h_i^s - h_j^s, \mathbf{1}_{ij}]) \quad (7) \qquad a_{ij}^s = \text{Sigmoid}(g(\tilde{h}_i^s - \tilde{h}_j^s)) \quad (9)$$

$$\tilde{h}_i^s = [h_i^s, u_i] \quad (8) \qquad h_i^{s+1} = \text{GRU}(h_i^s, \sum\nolimits_{j \in \mathcal{N}(i)} a_{ij}^s m_{ij}^s). \quad (10)$$

where $\mathcal{N}(i)$ is the set of node $i$ along with its neighboring nodes. $m_{ij}^s$ is the message sent from node $i$ to $j$ at the $s$-th message passing step. The connectivity for the propagation in GNN is given by $A_{1:i-1,1:i-1}$ with the last node (for which $A_{i,:}$ has not been generated yet) being fully connected. Note that message passing step is different from the generation step and we run multiple message passing steps per generation step in order to capture the structural dependency among nodes and edges. The $f$ and $g$ are two-layer MLPs. Since graphs are DAGs in our case rather than undirected ones as in (Liao et al., 2019), we add $\mathbf{1}_{ij}$ in Eq. (7), a one-hot vector for indicating the direction of the edge. We initialize the node representations $h_i^0$ (for $i < n$) as the corresponding one-hot encoded operation type vectors; $h_n^0$ is initialized to a special one-hot vector. Here $u_i$ is an additional feature vector that helps distinguish $i$-th node from others. We found using one-hot-encoded incoming neighbors of $i$-th node and a positional encoding of the node index $i$ work well in practice. We encourage readers to reference Fig. 4 for a detailed visualization of graph generation process.

**Sampling.** To sample from our generator, we first draw architectures following the standard ancestral sampling where each step involves drawing random samples from a categorical distribution and a mixture of Bernoulli distribution. At each step, this sampling process adds a new operator with a certain operation type and wire it to previously sampled operators.

### 3.1.2 EVALUATOR

Our design of generator and NAS pipeline do not rely on a specific choice of evaluator. Motivated by (Mnih et al., 2013), we use a replay buffer for storing the evaluated architectures. In our paper, based on specific datasets, we explore three types of evaluators, namely, oracle evaluator, supernet evaluator and benchmark evaluator, which are briefly introduced as follows.

**Oracle evaluator.** Given a sample from the generator, an oracle evaluator trains the corresponding network from scratch and tests it to get the validation performances. To reduce computation overhead, a common approach is to use early stopping (training with fewer epochs) as in (Tan et al., 2019; Tan & Le, 2019). In our experiment, we instead use a low-data evaluator similar to few-shot learning where we keep the same number of classes but use fewer samples per class to train.

**SuperNet evaluator.** Aiming at further reducing the amount of compute, this evaluator uses a weight-sharing strategy where each graph is a sub-graph of the supernet. We followed the single-path supernet setup used in (Pham et al., 2018) to compare with previous methods.

**Benchmark evaluator.** NAS benchmarks, *e.g.*, (Ying et al., 2019), provide accurate evaluation for architectures within the search space, which can be seen as oracle evaluators with full training budgets on target datasets.

### 3.2 LEARNING METHOD

Since we are dealing with discrete latent variables, *i.e.*, good architectures in our case, we train our NAS system using REINFORCE (Williams, 1992) algorithm with the control variate (a.k.a. baseline) to reduce the variance. In particular, the gradient of the loss or negative expected reward $\mathcal{L}$ w.r.t. the generator parameters $\phi$ is,

$$\nabla \mathcal{L}(\phi) = \mathbb{E}_{\mathbb{P}(\mathcal{G})} \left[ -\frac{\partial \log \mathbb{P}(\mathcal{G})}{\partial \phi} \bar{R}(\mathcal{G}) \right], \quad (11)$$

where the reward $\bar{R}$ is standardized as $\bar{R}(\mathcal{G}) = (R(\mathcal{G}) - C)/\sigma$. Here the baseline $C$ is the average reward of architectures in the replay buffer and $\sigma$ is standard deviation of rewards in the replay buffer. The expectation in Eq. (11) is approximated by the Monte Carlo estimation. However, the score function (*i.e.*, the gradient of log likelihood w.r.t. parameters) in the above equation may numerically differ a lot for different architectures. For example, if a negative sample, *i.e.*, an architecture with a reward lower than the baseline, has a low probability $\mathbb{P}(\mathcal{G})$, it would highly likely to have an extremely large absolute score function value, thus leading to a negative reward with an extremely large magnitude. Therefore, in order to balance positive and negative rewards, we propose to use the reweighted log likelihood as follows,

$$\log \mathbb{P}(\mathcal{G}) = \beta \mathbf{1}_{\bar{R}(\mathcal{G}) \leq 0} \log(1 - \mathbb{P}(\mathcal{G})) + \mathbf{1}_{\bar{R}(\mathcal{G}) > 0} \log(\mathbb{P}(\mathcal{G})) \tag{12}$$

where $\beta$ is a hyperparameter that controls the weighting between negative and positive rewards. $\mathbb{P}(\mathcal{G})$ is the original probability given by our generator.

**Exploration vs. Exploitation** Similar to many RL approaches, our NAS system faces the exploration vs. exploitation dilemma. We found that our NAS system may quickly collapse (*i.e.*, overly exploit) to a few good architectures due to the powerful graph generative model, thus losing the diversity and reducing to point estimate. Inspired by the epsilon greedy algorithm (Sutton & Barto, 2018) used in multi-armed bandit problems, we design a random explorer to encourage more exploration in the early stage. Specifically, at each search step, our generator samples from either itself or a prior graph distribution like the Watts–Strogatz model with a probability $\epsilon$. As the search goes on, $\epsilon$ is gradually annealed to 0 so that the generator gradually has more exploitation over exploration. Whats more, we design our replay buffer to keep a small portion of candidates. As training goes on, bad samples will be gradually be replaced by good samples for training our generators, which encourage the model to exploit more.

## 4 EXPERIMENTS

In this section, we extensively investigate our NAS system on three different search spaces to verify its effectiveness. First, we adopt the challenging RandWire search space (Xie et al., 2019) which is significantly larger than common ones. To the best of our knowledge, we are the first to explore learning NAS systems in this space. Then we search on the ENAS Macro (Pham et al., 2018) and NAS-Bench-101(Ying et al., 2019) search spaces to further compare with previous literature. For all experiments, we set the number of mixture Bernoulli $K$ to be 10, the number of message passing steps $S$ to 7, hidden sizes of node representation $h_i^s$ and message $m_{ij}^s$ to 128. For RNN-based baselines, we follow the design in (Zoph et al., 2018) if not other specified.

### 4.1 RANDWIRE SEARCH SPACE ON TINY-IMAGENET

**RandWire Search Space.** Originally proposed in (Xie et al., 2019), a randomly wired neural network is a ResNet-like four-stage network with the cell graph $\mathcal{G}$ defines the connectivity of $N$ convolution layers within each stage. At the end of each stage, the resolution is downsampled by $3 \times 3$ convolution with stride 2 whereas the number of channels is doubled. While following the RandWire small regime in (Xie et al., 2019), we share the cell graph $G$ among last three stages for simplification. To keep the number of parameters roughly the same, we fix the node type to be separable $3 \times 3$ convolution. The number of nodes $N$ within the cell graph $\mathcal{G}$ is set to 32 excluding the input and output nodes. This yields a search space of $2.1 \times 10^{149}$ valid adjacency matrices, which is extremely large and renders the neural architecture search challenging. More details of the RandWire search space can be found in the Appendix E.1.

**Tiny-ImageNet w. Oracle Evaluator.** To enable search on the RandWire space, we exploit the oracle evaluator on the Tiny-ImageNet dataset (Chrabaszcz et al., 2017). To save computation, we employ a low-data oracle evaluator where we sample $1/10$ of Tiny-ImageNet training set for training and use the rest for validation at each search step. Similar to the few-shot learning, we keep the number of classes unchanged but reduce the number of samples per class. After the search, we retrain our found architectures on the full training set and evaluate it on the original validation set. Specifically, for each model, the oracle evaluator trains for 300 epochs and uses the average validation accuracy of the last 3 epochs as the reward. Our total search budget is around 16 GPU days, which approximately amounts to 320 model evaluations, *e.g.*, 40 search steps and 8 samples evaluated per step. For random search baselines, we choose Erdős–Rényi (ER) and Watts–Strogatz (WS) models. Specifically, we first randomly draw hyperparameters from certain ranges, *i.e.*, $0.1 \leq$

| Methods | Cost (GPU Days) | Low Data (Search) | | Full Data (Final) | |
|---|---|---|---|---|---|
| | | Val Avg Acc | Std | Val Avg Acc | Std |
| ER-TopK | 15.2 | 23.12 | 0.34 | 61.76 | 0.04 |
| WS-TopK | 15.6 | 22.39 | 0.91 | 62.24 | 0.34 |
| ER-BEST | 15.2 | 20.07 | 1.62 | 62.10 | 0.25 |
| WS-BEST | 15.6 | 18.68 | 1.41 | 62.16 | 0.92 |
| RNN (Zoph et al., 2018) | 17.2 | 18.46 | **0.99** | 61.73 | 0.77 |
| Ours | 16.7 | **20.32** | 1.12 | **62.57** | **0.40** |

Table 1: Comparisons on Tiny-ImageNet. The top and bottom blocks include random search and learning-to-search methods respectively. ER-TopK and WS-TopK refers to top ($K$=4) architectures found by all WS and ER models during search. ER-BEST and WS-BEST refer to the best ER and WS models found during search, *i.e.*, WS($k$=4,$p$=0.75) and ER($p$=0.1). Here Avg and Std of accuracies are computed based on 4 architectures sampled from generators.

$p \leq 0.5$ for ER and $(2, 0.2) \leq (k, p) \leq (6, 0.8)$ for WS, and then sample $\mathcal{G}$ from individual models. We set the reweight coefficient $\beta$ to 0.05. For the random explorer, we choose WS model with the same hyperparameter range as a prior distribution and set $\epsilon = 0.6$ in the beginning and decay it by a factor of 0.2 every 10 search steps. We also find that gradually shrinking replay buffer size to keep 30% to 10% of top-performing architectures helps stabilize the training of the generator. At the search time, we reject samples that already appear in the replay buffer to avoid duplications. We apply the same setting to the RNN generator for a fair comparison.

**Results.** As shown in Table 1, we compare our NAS system with other random search methods and learning-to-search methods. We can see that our method outperforms the RNN-based generator and other random search methods in terms of average validation accuracy on the full dataset. Our generator also has a lower variance compared to the RNN-based one. Moreover, we observed that RNN based generator sometimes degenerates so that it frequently sample densely-connected graphs. This is probably due to the fact that RNN based generator does not effectively utilize the topology information. We can see that a high search reward (*i.e.*, a low-data validation accuracy) do not necessarily lead to better performances in full data training, which indicates a bias of the oracle evaluator within the low-data regime. Random search methods are prone to be biased as they select architectures solely based on the search reward. Nevertheless, our generator is less affected by the bias and able to learn a distribution of good architectures that perform well on full data training.

We also show results of the best architectures found within 4 samples in Table 2. Here, ER-top-1 and WS-top-1 refers to the best model found from corresponding random search. FC refers to the fully-connected graph, which takes three times longer to train compared to our model. It is clear that the best model found by our method outperforms those discovered by other methods by a considerable margin. Moreover, we scale up the best models (denoted as large) by adding more channels and one more computation stage (more details are in Appendix E.1). We

| Model | Param (M) | Top1 / Top5 Acc | |
|---|---|---|---|
| Resnet18 | 11.68 | $59.71_{\pm 0.09}$ | $80.32_{\pm 0.10}$ |
| Resnet50 | 25.56 | $63.42_{\pm 0.30}$ | $82.61_{\pm 0.15}$ |
| Resnext50 | 27.56 | $63.62_{\pm 0.07}$ | $82.73_{\pm 0.08}$ |
| FC | 3.49 | $60.82_{\pm 0.24}$ | $82.29_{\pm 0.09}$ |
| ER-Top1 | 3.23 | $61.82_{\pm 0.09}$ | $82.30_{\pm 0.18}$ |
| RS-Top1 | 3.22 | $62.55_{\pm 0.15}$ | $82.64_{\pm 0.21}$ |
| RNN | 3.32 | $62.29_{\pm 0.39}$ | $82.16_{\pm 0.24}$ |
| Ours | 3.27 | $\mathbf{63.23}_{\pm 0.18}$ | $\mathbf{83.06}_{\pm 0.05}$ |
| WS-Top1 Large | 19.38 | $63.84_{\pm 0.13}$ | $82.61_{\pm 0.16}$ |
| RNN Large | 19.78 | $63.69_{\pm 0.28}$ | $82.74_{\pm 0.21}$ |
| Ours Large | 19.18 | $\mathbf{64.45}_{\pm 0.26}$ | $\mathbf{83.23}_{\pm 0.26}$ |

Table 2: Comparisons of best searched architectures (averaged over 3 runs per architecture) on Tiny ImageNet.

can see that our searched architectures perform favorably against manually designed architectures like ResNet (He et al., 2016) and ResNeXt (Xie et al., 2017).

## 4.2 ENAS MACRO SEARCH SPACE ON CIFAR10

**ENAS Macro Search Space**, originally proposed by Pham et al. (2018), is a search space which focuses on the entire network. $\mathcal{G}$ here defines the entire network with $N = 12$ nodes. The operation type ($D$=6)[1] per node is also searchable. $\mathcal{G}$ is guaranteed to contain a length-11 path, *i.e.*, $\forall i > 1$, $A_{i,i-1} = 1$. The goal is to search off-diagonal entries, *i.e.*, skip connections. This gives a search space of $1.6 \times 10^{29}$ valid networks in total.

---

[1] $1 \times 1$, $5 \times 5$ convolution, $1 \times 1$, $5 \times 5$ separable convolution, max pooling, avg pooling

| Methods | Search Cost (days) | Params (M) | Best Error Rate | Top Samples Avg | Std |
|---|---|---|---|---|---|
| Net Transform (Cai et al., 2018) | 10 | 19.7 | 5.7 | - | - |
| NAS (Zoph & Le, 2016) | 22400 | 7.1 | 4.47 | - | - |
| PNAS (Liu et al., 2018a) | 225 | 3.2 | 3.41 | - | - |
| Lemonade (Elsken et al., 2018a) | 56 | 3.4 | 3.6 | - | - |
| EPNAS-Macro (Perez-Rua et al., 2018) | 1.2 | 38.8 | 4.01 | - | - |
| RNN* (Pham et al., 2018) | 0.9 | 19.64 | 4.18 | 4.47 | 0.282 |
| RNN* Large | 0.9 | 36.92 | 4.00 | 4.16 | 0.089 |
| Ours | 0.5 | 20.47 | 3.73 | 3.93 | 0.098 |
| Ours Large | 0.5 | 37.71 | **3.55** | **3.62** | **0.050** |

Table 3: Comparisons on CIFAR10 dataset. The bottom and top blocks include NAS methods with ENAS Macro and other search spaces respectively. *: our re-implementation. -: inapplicable.

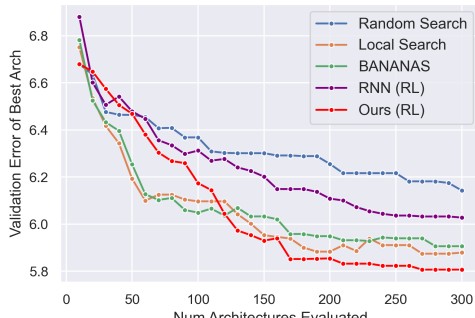

Figure 2: Performances (average over 10 runs) of best architectures vs. the number of architecture evaluations (search step).

| Method | Avg Error | #Queries |
|---|---|---|
| GCN Pred$^†$ | 6.331 | 150 |
| Evolution$^†$ | 6.109 | 150 |
| Ours | 5.930 $_{\pm 0.143}$ | 150 |
| Random Search | 6.413 $_{\pm 0.422}$ | 300 |
| Local Search | 5.879 $_{\pm 0.371}$ | 300 |
| BANANAS | 5.906 $_{\pm 0.296}$ | 300 |
| RNN (RL) | 6.028 $_{\pm 0.228}$ | 300 |
| Ours (RL) | **5.807** $_{\pm 0.072}$ | 300 |

Table 4: Best model performances on NAS-Bench-101. $^†$ indicates numbers are taken from (White et al., 2019) Table 2.

**SuperNet Evaluator.** For ENAS Macro search space, we experiment on CIFAR10 (Krizhevsky et al., 2009) dataset. For our generator, we use the ER model with $p = 0.4$ as our explorer, where $\epsilon$ decays from 1 to 0 in the first 100 search steps. For RNN based generator, we follow the setup in (Pham et al., 2018). We also adopt the weight-sharing mechanism in (Pham et al., 2018) to obtain a SuperNet evaluator that efficiently evaluates a model's performance. We use a budget of 300 search steps with around 100 architectures evaluated per step for all methods. After the search, we use a short-training of 100 epochs to evaluate the performances of 8 sampled architectures, after which top-4 performing ones are chosen for a 600-epoch full training. The best validation error rate among these 4 architectures is reported. For simplicity and a fair comparison, we do not use additional tricks (*e.g.*, adding entropy regularizer) in (Pham et al., 2018). More details are provided in Appendix F.

In Table 3, we compare the error rates and variances for different NAS methods. Note that this variance reflects the uncertainty of the distribution of architectures as it is computed based on sampled architectures. It is clear that our GraphPNAS achieves both lower error rates and lower variances compared to RNN based generator and is on par with the state-of-the-art NAS methods on other search spaces. We also see that the best architecture performance of our generator outperforms RNN based generator by a significant margin. This verifies that our GraphPNAS is able to learn a distribution of well-performing neural architectures. Given that we only sample 8 architectures, the performances could be further improved with more computational budgets.

### 4.3 NAS Benchmarks

**NAS-Bench-101** (Ying et al., 2019) is a tabulate benchmark containing 423K cell graphs, each of which is a DAG with up to 7 nodes and 9 edges including input and output nodes. We compare the performances of our GraphPNAS to open-source implementations of random search methods, local search methods, and BANANAS (White et al., 2019). The latter two are the best algorithms found by White et al. (2020) on NAS-Bench-101. For GCN prediction and evolution methods, we use the score reported in (White et al., 2020). We give each NAS method the same budget of 300 queries and plot the curve of lowest test error as a function of the number of evaluated architectures. As shown in Fig. 2, our GraphPNAS is able to quickly find well-performing architectures. We also report the avg error rate over 10 runs in Table 4. Our GraphPNAS again outperforms RNN

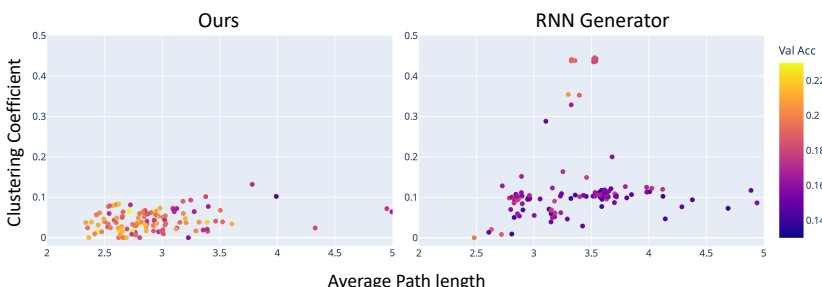

Figure 3: Visualization of architecutre explore sapce of GraphPNAS vs RNN. Each point in the figure denotes a model evaluation. Colors of each node denotes its validation accuracy returned by the low-data Oracle evaluator.

based generator by a significant margin and beats strong baselines like local search methods and BANANAS. Notably, our GraphPNAS has a much lower variance than other methods, thus being more stable across multiple runs.

**NAS-Bench-201** (Dong & Yang, 2020) is defined on a smaller search space where up to 4 nodes and 6 edges are allowed. Experimental results can be found in Appendix A.

## 5 DISCUSSION & CONCLUSION

**Qualitative Comparisons between RNN and GraphPNAS.** In (You et al., 2020), the clustering coefficient and the average path length have been used to investigate distributions of graphs. Here we adopt the same metrics to visualize architectures (graphs) sampled by RNN based and our generators in RandWire experiments. Points in Fig. 3 refer to architectures sampled from both generators in the last 15 search steps where random explorers are disabled. The validation performances are color-coded. We can see that our GraphPNAS samples a set of graphs that have better validation accuracies while the ones of RNN generator have large variances in performances. Moreover, the graphs in our case concentrate to those with smaller clustering coefficients, thus less likely being densely-connected. On the contrary, RNN generator tends to sample graphs that are more likely to be densely-connected. While RNN has been widely used for NAS, we show in our experiments that our graph generator consistently outperforms RNN over three search spaces on two different datasets. This is likely due to the fact that our graph generator better leverages graph topologies, thus being more expressive in learning the distribution of graphs.

**Bias in Evaluator.** In our experiments, we use SuperNet evaluator, low-data, and full-data Oracle evaluator to efficiently evaluate the model. From the computational efficiency perspective, one would prefer the SuperNet evaluator. However, it tends to give high rewards to those architectures used for training SuperNet. Although the low-data evaluator is more efficient than the full-data one, its reward is biased as discussed in Section 4.1. This bias is caused by the discrepancy between the data distributions in low-data and full-data regimes. We also tried to follow (Tan et al., 2019) to use early stopping to reduce the time cost of the full-data evaluator. However, we found that it assigns higher rewards to those shallow networks which converge much faster in the early stage of training. We show detailed results in Appendix E.4.

**Search Space Design.** The design of search space largely affects the performances of NAS methods. Our GraphPNAS successfully learns good architectures on the challenging RandWire search space. However, the search space is still limited as the cell graph across different stages is shared. A promising direction is to learn to generate graphs in a hierarchical fashion. For example, one can first generate a macro graph and then generate individual cell graphs (each cell is a node in the macro graph) conditioned on the macro graph. This will significantly enrich the search space by including the macro graph and untying cell graphs.

**Conclusion.** In this paper, we propose a GNN-based graph generator for NAS, called GraphP-NAS. Our graph generator naturally captures topologies and dependencies of operations of well-performing neural architectures. It can be learned efficiently through reinforcement learning. We extensively study its performances on the challenging RandWire as well as two widely used search spaces. Experimental results show that our GraphPNAS consistently outperforms RNN-based generator on all datasets. Future works include exploring ensemble methods based on our GraphPNAS and hierarchical graph generation on even larger search spaces.

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

## A  EXPERIMENTS ON THE NAS-BENCH-201 SEARCH SPACE

Here we compare our method on NAS-Bench-201(Dong & Yang, 2020) with random search (RAN-DOM) Bergstra & Bengio (2012), random search with parameter sharing (RSPS) Li & Talwalkar (2020), REA Real et al. (2019b), REINFORCE Williams (1992), ENAS Pham et al. (2018), first order DARTS (DARTS 1st) Liu et al. (2018b), second order DARTS (DARTS 2nd), GDAS Dong & Yang (2019c), SETN Dong & Yang (2019b), TAS Dong & Yang (2019a), FBNet-V2 Wan et al. (2020), TuNAS Bender et al. (2020), BOHB Falkner et al. (2018).

| Methods | CIFAR-10 | | CIFAR-100 | | ImageNet-16-120 | |
|---|---|---|---|---|---|---|
| | validation | test | validation | test | validation | test |
| *Weight Sharing NAS:* | | | | | | |
| RSPS | 87.60±0.61 | 91.05±0.66 | 68.27±0.72 | 68.26±0.96 | 39.73±0.34 | 40.69±0.36 |
| DARTS (1st) | 49.27±13.44 | 59.84±7.84 | 61.08±4.37 | 61.26±4.43 | 38.07±2.90 | 37.88±2.91 |
| DARTS (2nd) | 58.78±13.44 | 65.38±7.84 | 59.48±5.13 | 60.49±4.95 | 37.56±7.10 | 36.79±7.59 |
| GDAS | 89.68±0.72 | 93.23±0.58 | 68.35±2.71 | 68.17±2.50 | 39.55±0.00 | 39.40±0.00 |
| SETN | 90.00±0.97 | 92.72±0.73 | 69.19±1.42 | 69.36±1.72 | 39.77±0.33 | 39.51±0.33 |
| ENAS | 90.20±0.00 | 93.76±0.00 | 70.21±0.71 | 70.67±0.62 | 40.78±0.00 | 41.44±0.00 |
| *Multi-trial NAS:* | | | | | | |
| REA | 91.25±0.31 | 94.02±0.31 | 72.28±0.95 | 72.23±0.84 | 45.71±0.77 | **45.77±0.80** |
| REINFORCE | 91.12±0.25 | 93.90±0.26 | 71.80±0.94 | 71.86±0.89 | 45.37±0.74 | 45.64±0.78 |
| RANDOM | 91.07±0.26 | 93.86±0.23 | 71.46±0.97 | 71.55±0.97 | 45.03±0.91 | 45.28±0.97 |
| BOHB | 91.17±0.27 | 93.94±0.28 | 72.04±0.93 | 72.00±0.86 | **45.55±0.79** | 45.70±0.86 |
| Ours | **91.47±0.15** | **94.28±0.17** | **72.47±0.84** | **72.34±0.81** | 45.00±0.83 | 45.40±0.97 |

Table 5: Searched best architecture performance on NAS-Bench-201. We run our methods 10 times to obtain mean and standard deviation.

To fairly compare with scores reported in (Dong et al., 2021), we fix a search budget of 20000s on CIFAR10 and CIFAR100, and 30000s on ImangeNet-16-120, which is approximately 150, 80, and 40 oracle evaluations (with 1 sample evaluated per step) on CIFAR10, CIFAR100, and ImageNet-16-120 respectively. Specifically for NAS-Bench-201, we use random explorer in the first 10 steps and keep the top 15 architectures in the replay buffer. We found that our model outperforms previous methods on CIFAR10 and CIFAR100 datasets and is on par with state-of-the-art methods on ImageNet-16-120 dataset. **On ImageNet-16-120, after we extend the number of search budget from 40 to 60 steps, we significantly boost the performance to 45.57(+0.57) and 45.79(+0.4) on validation and test sets respectively.** This indicates that the search process of our model hasn't converged due to the limited search steps. This suggests that a reasonable number of search steps is needed for our model to reach its full potential.

## B  COMPARISON WITH NAGO

Following Xie et al. (2019)'s work on random graph models, Ru et al. (2020) propose to learn parameters of random graph models using bayesian optimization. We compare with the randwire search space (refers to as RANG) in (Ru et al., 2020). Since the original search space in (Xie et al., 2019) do not reuse cell graphs for different stages, we train conditionally independent graph generators for different stages respectively. That is 3 conditionally independent generators for $conv_3$, $conv_4$, and $conv_5$ stage in Table 9. We perform a search on the CIFAR10 dataset, where each model is evaluated for 100 epochs. We restrict the search budget to 600 oracle evaluations. We align with settings in (Ru et al., 2020) for retraining and report sampled architecture's test accuracy and standard deviation in the Table 6. We can see that our method learns a distribution of graphs that outperforms previous methods.

## C  EXPERIMENTS ON THE DARTS SEARCH SPACE

DARTS is a small cell-graph-based search space (Liu et al., 2018b). It is defined on a DAG with up to 6 nodes and 8 possible operators. Each node can have no more than two inputs. Like NAS-Bench-201, node operations are defined on the edge where nodes represent fuse operation between

| Methods | Reference | Avg. Test Accuracy (%) | Std. |
|---|---|---|---|
| RANG-D | Xie et al. (2019) | 94.1 | 0.16 |
| RANG-BOHB | Ru et al. (2020) | 94.0 | 0.26 |
| RANG-MOBO | Ru et al. (2020) | 94.3 | 0.13 |
| Ours | - | **94.6** | 0.18 |

Table 6: Comparison of the searched results on CIFAR10. Mean test accuracy and the standard deviation are calculated over 8 samples from the searched generator. We align the search space design and retraining setting for a fair comparison.

incoming edges. Desipte the fact that NAS performances on CIFAR10 with DARTS is already saturated, we experiment on it to compare with previous NAS methods. To be consistent with DARTS, we split CIFAR10 into subsets containing 40000 training and 10000 validation samples and train architectures using SGD for 50 epochs with an initial learning rate of 0.025 (annealed to 0 via cosine decay), momentum of 0.9, weight decay of 0.0003, as well as auxiliary towers, drop path, and cutout. We allow a budget of 96 oracle evaluations, or roughly 72 hours on 4 NVIDIA TITAN Xp GPUs. We follow the DARTS final evaluation pipeline and perform SGD for 600 epochs with an initial learning rate 0.025 annealed to 0 by cosine decay, momentum 0.9, and weight decay 0.0003. We also perform data augmentation (cropping/horizontal flipping) and adapt auxiliary towers, drop path, and cutout.

| Architecture | Reference | Test Error (%) | Params (M) | Search Cost (GPU hr.) |
|---|---|---|---|---|
| ENAS | Pham et al. (2018) | 2.89 | 4.6 | 11 |
| NSGA-NET | Lu et al. (2019) | 2.75 | 3.3 | 7 |
| DARTS | Liu et al. (2018b) | 2.76 | 3.3 | 96 |
| PDARTS | Chen et al. (2019) | 2.50 | 3.4 | 7 |
| RF-DARTS | Zhang et al. (2020) | 3.65 | - | 4.3 |
| PCDARTS | Xu et al. (2020) | 2.57 | 3.4 | - |
| DARTS- | Chu et al. (2020) | 2.58 | - | 3.5 |
| DUDARTS | Lu et al. (2021) | **2.38** | 3.7 | 0.4 |
| BANANAS | White et al. (2019) | 2.64 | - | 283 |
| Ours | - | 2.77 | 3.4 | 288 |

Table 7: Comparison of results on CIFAR10 with DARTS search space.

From Table 7, we can see that our method is on par with the state-of-art methods. The computation cost is majorly due to the fact that oracle evaluation is used, where future works using a performance predictor or SuperNet-based evaluator could be exploited to reduce the computation. Due to limits on available computation, we reuse the best hyperparameters found in NAS-bench-101. We expect further hyperparameter tuning would further boost the performance. We also provide search results on CIFAR100. Due to time constraints, we only compare with a few open-source NAS methods. For a fair comparison, we reuse the final training hyperparameters from Chen et al. (2019).

| Architecture | Test Error (%) | Params (M) | Search Cost (GPU hr.) |
|---|---|---|---|
| DARTS | 19.24 | 3.3 | 96 |
| PDARTS (trained on CIFAR-100)* | 15.92 | 3.6 | 0.3 |
| PDARTS (trained on CIFAR-10)* | 16.55 | 3.4 | 0.3 |
| Random Sampling | 17.57 | 3.3 | – |
| Random Search | 16.70 | 3.4 | 288 |
| Ours | **15.89** | 3.4 | 288 |

Table 8: Comparison of results on CIFAR100 with DARTS search space.

## D    MORE DETAILS ON GRAPH GENERATOR

To more clearly illustrate the sampling process of our generator, we detailed probabilistic sampling process of our generator in Fig. 4.

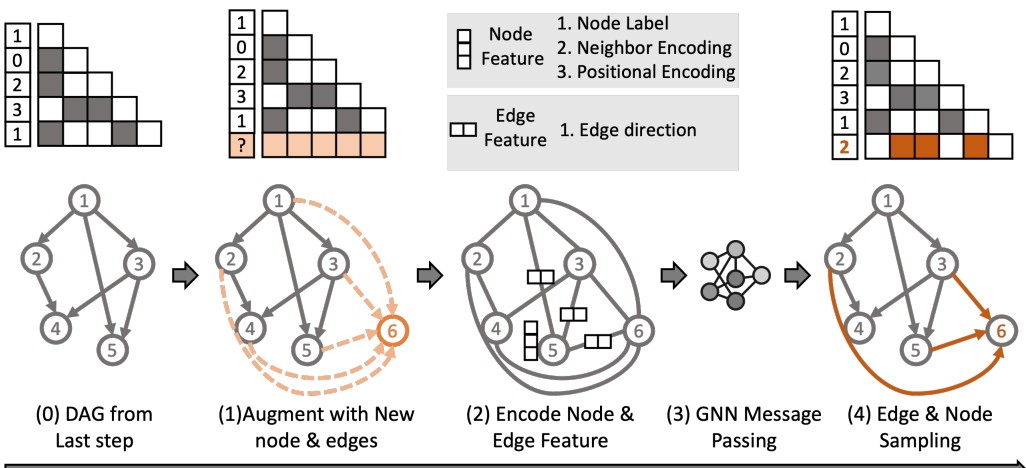

Figure 4: Detailed steps of auto-regressive generation with our graph generator.

# E  MORE DETAILS ON THE RANDWIRE EXPERIMENTS

## E.1  DETAILS OF RANDWIRE SEARCH SPACE

Here we provided more details on the RandWire search space shown in Table 9 and Fig. 5.

| Stage | Output | Base | | Large | |
|---|---|---|---|---|---|
| | | Cell | Channels | Cell | Channels |
| conv$_1$ | 112×112 | conv$_{3×3}$ | 32 | conv$_{3×3}$ | 48 |
| conv$_2$ | 56×56 | conv$_{3×3}$ | 64 | conv$_{3×3}$ | 96 |
| conv$_3$ | 28×28 | $\mathcal{G}$ | 64 | $\mathcal{G}$ | 192 |
| conv$_4$ | 14×14 | $\mathcal{G}$ | 128 | $\mathcal{G}$ | 288 |
| | | | | $\mathcal{G}$ | 384 |
| conv$_5$ | 7×7 | $\mathcal{G}$ | 256 | $\mathcal{G}$ | 586 |
| classifier | 1×1 | 1×1 conv$_{1×1}$, 1280-d | | | |
| | | global average pool, 200-d $fc$, softmax | | | |

Table 9: Randwire search space with base and large settings. Base is the default setting for search while Large refers to the architecture of scaled up models in Table 2. conv denote a ReLU-SepConv-BN triplet . The input size is 224×224 pixels. The change of the output size implies a stride of 2 (omitted in table) in the convolutions that are placed at the end of each block. $\mathcal{G}$ is the shared cell graph that has $N = 32$ node.

## E.2  DETAILS FOR RANDWIRE EXPERIMENTS

For experiment on Tiny-Imagenet, we resize image to $224 \times 224$ as showed in Table 9. We apply the basic data augmentation of horizontal random flip and random cropping with padding size 4. We provide detailed hyper-parameters for oracle evaluator training and learning for GraphPNAS in Table 10

## E.3  VISUALIZATION OF ARCHITECTURES FROM OUR GENERATOR

Here we visualize the top candidate architectures in Fig. 6

## E.4  BIAS FOR EARLY STOPPING

As discussed in Section 5, using early stopping will lead to local minimal where the generator learns to generate shallow cell structure. We quantify this phenomenon in table 11, where we can see that

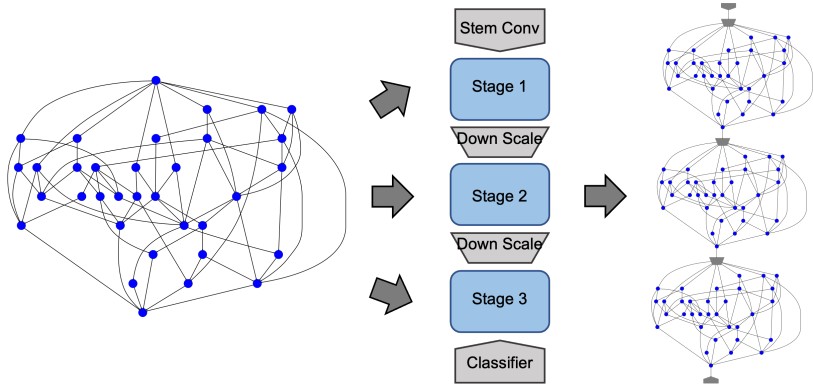

Figure 5: Visualization of RandWire search base space used in this paper. Different from (Xie et al., 2019), $\mathcal{G}$ here is shared across three stages.

| Oracle Evaluator | | Graph Controller | |
|---|---|---|---|
| batch size | 256 | graph batch size | 16 |
| optimizer | SGD | generator optimizer | Adam |
| learning rate | 0.1 | generator Learning rate | 1e-4 |
| learning rate deacy | consine lr decay | generator learning rate decay | none |
| weight decay | 1e-4 | generator weight decay | 0. |
| grad clip | 0. | generator gradient clip | 1.0 |
| training epochs | 300 | replay buffer fitting epochs | 2000 |

Table 10: Hyperparameter setting for oracle evaluator and training our graph generator.

with early stopping training, the generator will generate more shallow architectures with a shorter path from input to output. The corresponding average final validation accuracy also dropped by a large margin compared to the low data evaluator counter part.

| evaluator | Final Val Acc | Average Path | Longest Path |
|---|---|---|---|
| early stopping | 61.86 | 2.595 | 6.125 |
| low data regime | 62.57 | 3.046 | 8.75 |

Table 11: In the table, we show ablation on the choice of oracle evaluator with our graph generator. The average Path and Longest path are computed as the average path length and longest path length from input to output over 8 samples from the corresponding generator.

## F   MORE DETAILS ON ENAS MACRO EXPERIMENTS

For ENAS Macro search space, we use a pytorch-based open source implementation[2] and follow the detailed parameters provided in (Pham et al., 2018) for RNN generator. Specifically, we follow (Pham et al., 2018) to train the SuperNet and update the generator in an iterative style. At each search step, two sets of samples $\mathcal{G}_{\text{train}}$ and $\mathcal{G}_{\text{eval}}$ are sampled from the generator. $\mathcal{G}_{\text{train}}$ is used to learn the SuperNet's weights by back-propagating the training loss. The updated SuperNet is used for evaluating $\mathcal{G}_{\text{eval}}$, which is then used for updating the generator.

For our generator, we evaluate 100 architectures per step and update our generator every 5 epochs of SuperNet training. Instead of evaluating on a single batch, we reduce the number of models evaluated per step and evaluate on the full test set. We found this stables the training of our generator while keeping evaluation costs the same. In the replay buffer, the top 20% of architectures is kept.

For training SuperNet and RNN generator, we follow the same hyper-parameter setting in (Pham et al., 2018) except the learning rate decay policy is changed to cosine learning rate decay. For

---

[2]https://github.com/microsoft/nni/tree/v1.6/examples/nas/enas

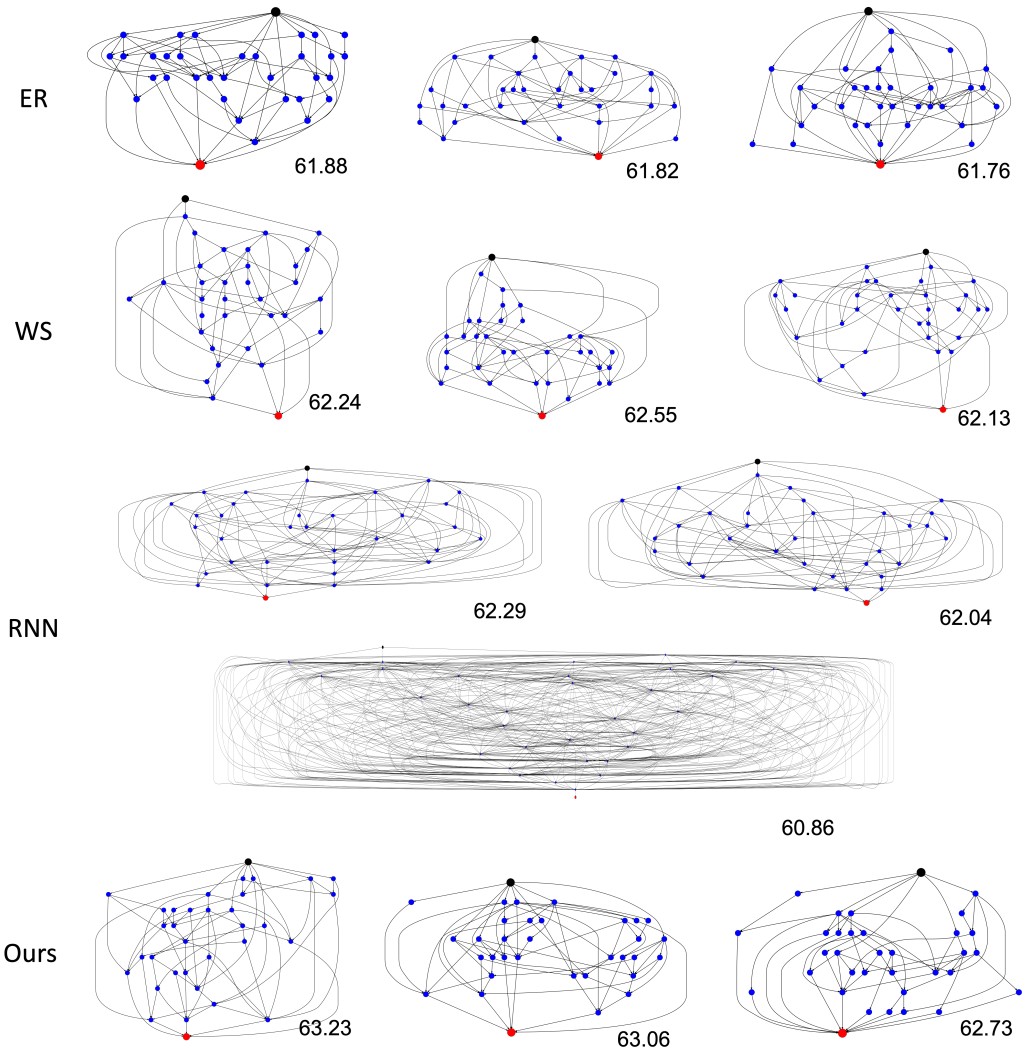

Figure 6: Visualization of Top 3 architectures sampled by each method. We observe that around 50% of samples from RNN generators are densely connected graphs or even fully connected graphs.

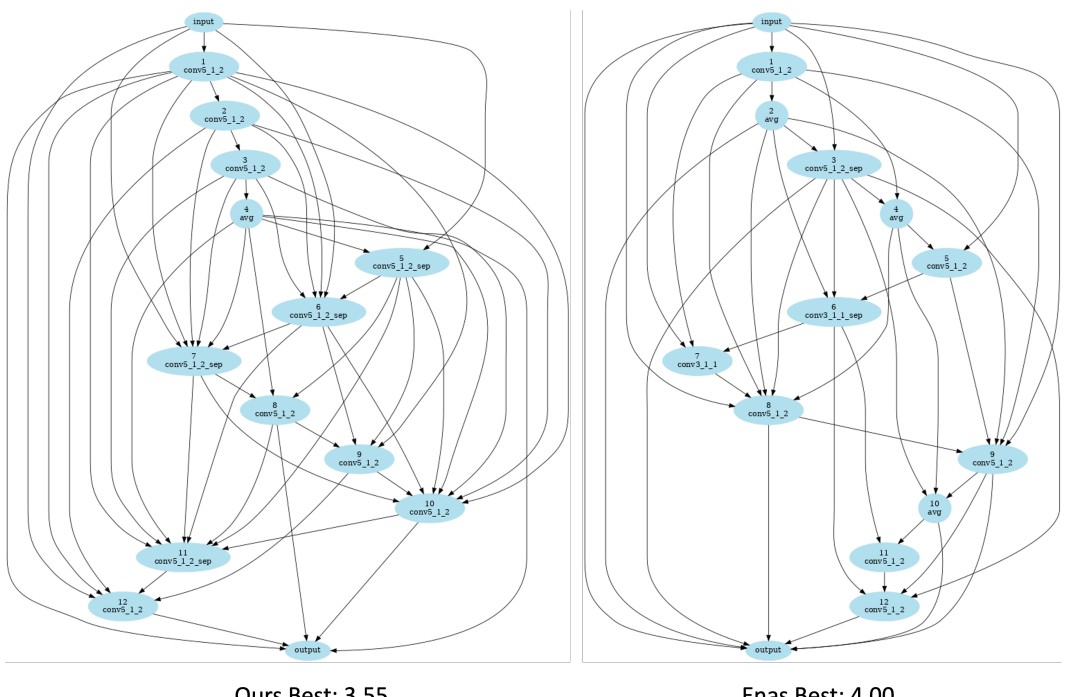

Ours Best: 3.55         Enas Best: 4.00

Figure 7: The best architecture found by GraphPNAS and RNN generator (Pham et al., 2018). Correspond to scores report in table 3. To get this architecture, we pre-evaluate 8 samples for both methods and select top-performing architecture.

training our generator we use the same hyperparameter as in Table 10 with graph batch size changed to 32. For retraining the found the best architecture, we use a budget of 600 epoch training with a learning rate of 0.1, batch size 256, and weight decay 2e-4. We also apply a cutout with a probability of 0.4 to all the models when retraining the model.

### F.1 VISUALIZATION OF BEST ARCHITECTURE FOUND

Here we visualize the best architecture found by GraphPNAS and RNN generator for Enas Macro search space in Fig. 7.

## G MORE DETAILS ON NAS-BENCH-101

For sampling on NAS-Bench-101 (Ying et al., 2019), we first sample a 7-node DAG, then we remove any node that is not connected to the input or the output node. We reject samples that have more than 9 edges or don't have a path from input to output.

To train our generator on Nas-Bench-101, we use Erdős–Rényi with $p = 0.25$, $\epsilon$ is set to 1 in the beginning and decreased to 0 after 30 search steps. For the replay buffer, we keep the top 30 architectures. Our model is updated every 10 model evaluations, where we train 70 epochs on the replay buffer at each update time. The learning rate is set to 1e-3 for with a batch size of 2 on Nas-bench-101. per

