# OpenReview forum: "GraphPNAS: Learning Distribution of Good Neural Architectures via Deep Graph Generative Models"
_ICLR.cc/2023/Conference — Submitted to ICLR 2023_

### Official Review · Reviewer_QTZh · 2022-10-23

**Confidence:** 3
**Correctness:** 3
**Technical Novelty And Significance:** 3
**Empirical Novelty And Significance:** 3
**Recommendation:** 5

**Clarity, Quality, Novelty And Reproducibility:**

This paper proposed a clear framework, and the idea is easy to follow. The extensive experimental results on several benchmark datasets demonstrated the effectiveness of the proposed method. However, the lacking of open-source codes may limit reproducibility.

**Strength And Weaknesses:**

Strength:
1. The idea of considering the NAS as GNN is interesting.
2. The paper is well-organized and easy to follow.


Weaknesses:
1. The experiment does not show the efficiency of the method. Would you consider adding computational complexity analysis or time consumption?
2. I don't quite understand why this method is less effective than other methods when the number of architectures evaluation is small. It suddenly got better as the number increased （Figure 2). Can you explain?
3. In addition, the authors do not conduct experiments on the most common DARTS space, making it hard to conduct comparisons with most existing SOTA NAS methods. I understand that the DARTS space is an old benchmark and not easy to get significant improvement, while I am still interested in the performance of the proposed method in this space.

**Summary Of The Paper:**

This paper proposes a probabilistic graph generator that models the distribution over good architectures using GNNs. This converts the neural architectures to a learnable computational graph. Extensive experiments indicate the effectiveness and efficiency of the proposed method. This paper is generally well-written, and the idea is interesting.

**Summary Of The Review:**

This paper is generally well-written, and the idea is novel. I'd like to increase the score if the authors can resolve the above concerns.

---

### Official Review · Reviewer_kb9c · 2022-10-23

**Confidence:** 3
**Correctness:** 3
**Technical Novelty And Significance:** 2
**Empirical Novelty And Significance:** 2
**Recommendation:** 5

**Clarity, Quality, Novelty And Reproducibility:**

The paper is clearly written. Not reproducible as code is not provided and it is not trivial to implement the proposed method.

**Strength And Weaknesses:**

Strength:
1. Some novelty as the first paper to propose using GNN-based generative modeling for generating architectures in NAS.
2. Improved results compared to an RNN-based generator and several baselines that are quite old though.

Weakness:
1. I feel the cited papers and compared baselines are quite out-of-date. Cited works are mostly before 2020. While I am not well literatured in NAS, I do know there were multiple NAS methods proposed in 2021/2022. Maybe most of them are differentiable NAS methods but some comparisons should be made.
2. Can evaluation be performed on the larger ImageNet and the newer NAS-bench-201 datasets? I feel results on the smaller and older datasets, without many more recent baselines, are less convincing.

**Summary Of The Paper:**

The paper proposed to use GNN as the architecture controller in reinforcement learning based NAS.

**Summary Of The Review:**

An interesting paper that first proposed using GNN as architecture controller, but need to be benchmarked with more recent methods and on more datasets.

---

### Official Review · Reviewer_DyTs · 2022-10-24

**Confidence:** 3
**Correctness:** 3
**Technical Novelty And Significance:** 3
**Empirical Novelty And Significance:** 3
**Recommendation:** 6

**Clarity, Quality, Novelty And Reproducibility:**

The writing is clear, the quality is good, and it is an interesting application of GNN.

**Strength And Weaknesses:**

### Strengths
1. The paper proposes a GNN-based graph generator for NAS, which is an interesting exploration.
2. The paper applies the method to extensive experiments and shows state-of-the-art results on multiple benchmarks.
3. The paper also introduces a larger search space and proves the efficiency of the method against the existing work.


### Weaknesses
1. More discussions of how the neural architecture is encoded in a GNN model would be appreciated. For example, figure 4 in the appendix should be in the main paper and illustrated.
2. The evaluation of the method is applied to smaller datasets (Tiny-Imagenet, which is not common in the previous work. It would be more interesting to see how the method performs on large-scale real applications.
3. It would be interesting to see different constraints for the NAS method, such as latency constraints of the neural architecture.

**Summary Of The Paper:**

The paper proposes a GNN-based neural architecture method that can model the distribution of well-performing architectures. With a reinforcement learning formulation, the paper claims that the learned GNN-generator is more flexible and efficient than existing work.

**Summary Of The Review:**

Please refer to above.

---

> ### Author Response · Authors · 2022-11-19
> **Response to Reviewer DyTs**
>
> We thank the reviewer for the positive and helpful comments!
>
> > **Q1: More discussions of how the neural architecture is encoded in a GNN model would be appreciated. For example, figure 4 in the appendix should be in the main paper and illustrated.**
>
> A1: Thanks for your good point! We moved Figure 4 to the main paper and added more discussion in the revision.
>
> > **Q2: The evaluation of the method is applied to smaller datasets (Tiny-Imagenet, which is not common in the previous work. It would be more interesting to see how the method performs on large-scale real applications.**
>
> A2: We would like clarify that previous methods mostly search on CIFAR10, and the improvement on CIFAR10 saturates. So we choose Tiny-ImageNet (which is 2x larger in size and 4x larger in resolution compared to CIFAR10) as it is more challenging. To search on ImageNet with reinforcement learning and oracle evaluations, one would need an enormous amount of computation (around 288 TPU days in [1,2]), which we cannot afford at this stage. Other approaches include using weight sharing mechanism and training-free evaluations. Enabling weight sharing or apply training-free evaluations on RandWire search space is not straight forward but will be interesting future directions.
>
> [1] Tan, Mingxing, et al. "Mnasnet: Platform-aware neural architecture search for mobile." Proceedings of the IEEE/CVF Conference on Computer Vision and Pattern Recognition. 2019.
>
> [2] Tan, Mingxing, and Quoc Le. "Efficientnet: Rethinking model scaling for convolutional neural networks." International conference on machine learning. PMLR, 2019.

---

### Official Review · Reviewer_8CC8 · 2022-10-25

**Confidence:** 5
**Correctness:** 3
**Technical Novelty And Significance:** 1
**Empirical Novelty And Significance:** 2
**Recommendation:** 3

**Clarity, Quality, Novelty And Reproducibility:**

Clarity: good
Quality: fair
Novelty: low
Reproducibility: good


**Strength And Weaknesses:**

Pros:
1.	The paper is clear written and easy to follow.
2.	The Figure3 show the architecture space model by the proposed generator is better than classical RNN controller.
3.	The authors use several different search spaces on different datasets to demonstrate model effectiveness.

Cons:
1.	The literature review is not sufficient. There are lots of papers use graph generators for NAS[1~3]. So this point is not new. The authors should explain how their model is more brilliant than these methods.
2.	The graph generator model designs are borrowed from previous works. There are few contributions in this part.
3.	More NAS baselines should be compared in the experiment part. The authors should also show why the architecture space model by the proposed graph generator is better than previous works.


[1]Zhang, Miao, et al. "Differentiable neural architecture search in equivalent space with exploration enhancement." Advances in Neural Information Processing Systems 33 (2020): 13341-13351.
[2]Shi, Han, et al. "Bridging the gap between sample-based and one-shot neural architecture search with bonas." Advances in Neural Information Processing Systems 33 (2020): 1808-1819.
[3]Ru, Robin, Pedro Esperanca, and Fabio Maria Carlucci. "Neural architecture generator optimization." Advances in Neural Information Processing Systems 33 (2020): 12057-12069.


**Summary Of The Paper:**

This paper propose a NAS algorithm named GraphPNAS, which uses a graph generator instead of classical RNN generator to sample architectures from the space. RL algorithm is used to update the generator. Experiments on several different search space show the effectiveness of the method.

**Summary Of The Review:**

This paper propose a NAS algorithm named GraphPNAS, which uses a graph generator instead of classical RNN generator to sample architectures from the space. However, the contributions are limited on model design. Furthermore, extend explanations and experiments are needed to support the method.

---

> ### Author Response · Authors · 2022-11-19
> **Response to Reviewer 8CC8**
>
>
> We thank the reviewer for the helpful comments!
>
> > Q1. **The literature review is not sufficient. There are lots of papers that use graph generators for NAS[1~3].**
>
> A1: Thanks for pointing out the relevant literature. We cited and discussed these works in th e revised paper.
>
> While these references are related to our work (e.g., using GNNs), our methods are different in that we apply a deep graph generative model to NAS where a graph is sampled from noises in an auto-regressive manner. [1] uses a graph VAE to encode the neural architecture and do gradient-based optimization. [2] uses GCN as a feature extractor for neural architectures. In a nutshell, both [1] and [2] do not build graph generative models.
>
> [3] proposes a hierarchical search space with a graph generator based on random graph models like WS and ER. Bayesian optimization is used to optimize the hyper-parameters of the random graph models. In contrast, we build GNN-based deep graph generative models which are more expressive than vanilla random graph models. Furthermore, we add comparisons of our methods and [3] based on the search space designed in [4]. Results can be found in Table 6 (also copied below) in Appendix B. Our method outperforms [3,4] on CIFAR10 which matches our expectation w.r.t. the expressiveness.
>
> | Methods  | Cifar10 Acc  | Std |
> |:----------|:----------|:----------|
> | RANG-D[4]    | 94.1   | 0.16 |
> | RANG-BOHB[3]    | 94.0   | 0.26 |
> | RANG-MOBO[3]    | 94.3   | **0.13** |
> | Ours    		  | **94.6**    | 0.18 |
>
>
> > **Q2 There are few contributions in the graph generator part**
>
> A2: We would like to clarify that our main contribution is to explore the potential of graph generators in NAS and then design a well-performing NAS system rather than building novel and or better graph generators, since the latter falls under an independent research area, i.e., deep generative models of graphs.
>
> > **Q3: More NAS baselines should be compared in the experiment part. The authors should also show why the architecture space model by the proposed graph generator is better than previous works.**
>
> A3: Thanks for the suggestion. We add comparisons with more recent baselines on two commonly used search spaces, i.e., NAS-Ben-201 (Table 5 in Appendix A) and DARTS (Table 7 and 8 in Appendix C). Results show that our method either outperforms or is on par with the state-of-the-art methods.
>
> The quality of learned probabilitic distribution of architectures can be quantified using mean accuracy and and standard deviation of performances of sampled architectures. We showed in Table 1 and Table 3 that our trained generator is able to sample architectures with higher mean and lower standard deviation of performances. We also show, qualitatively, in Figure 3, that our generator learns to sample a clustering of high quality architectures while RNN based generator fails to converge to good architectures.
>
> [1]Zhang, Miao, et al. "Differentiable neural architecture search in equivalent space with exploration enhancement." Advances in Neural Information Processing Systems 33 (2020): 13341-13351.
>
> [2]Shi, Han, et al. "Bridging the gap between sample-based and one-shot neural architecture search with bonas." Advances in Neural Information Processing Systems 33 (2020): 1808-1819.
>
> [3]Ru, Robin, Pedro Esperanca, and Fabio Maria Carlucci. "Neural architecture generator optimization." Advances in Neural Information Processing Systems 33 (2020): 12057-12069.
>
> [4]Xie, Saining, et al. "Exploring randomly wired neural networks for image recognition." Proceedings of the IEEE/CVF International Conference on Computer Vision. 2019.

---

### Decision · Program_Chairs · 2023-01-20

**Decision:**

Reject

**Justification For Why Not Higher Score:**

The authors provided point to point answers and added experiment results as permitted by the time of the response period. However, there still seems to be a lack of enough enthusiasm among the reviewers after the author responses.

**Justification For Why Not Lower Score:**

N/A

**Metareview: Summary, Strengths And Weaknesses:**

This paper proposed a NAS algorithm named GraphPNAS, which uses a probabilistic graph generator as the architecture controller in reinforcement learning based NAS. RL algorithm is used to update the generator. Experiments on several different search space show the effectiveness of the method. The paper claims that the learned GNN-generator is more flexible and efficient than existing work. The reviewers raised concerns on the added values of the proposed framework over existing graph generators for NAS, differentiations over similar graph generator model designs, more up-to-date NAS baselines, larger experimental datasets. The authors provided point to point answers and added experiment results as permitted by the time of the response period. However, there still seems to be a lack of enough enthusiasm among the reviewers after the author responses.